# Augmented Reality Lights for Compromised Visibility Navigation

**Doupadi Bandara [1,*], Michael Woodward [1], Christopher Chin [1] and Danchi Jiang [2]**

[1] Australian Maritime College, University of Tasmania, Launceston 7250, TAS, Australia; michael.woodward@utas.edu.au (M.W.); c.chin@utas.edu.au (C.C.)

[2] School of Engineering, University of Tasmania, Hobart 7005, TAS, Australia; danchi.jiang@utas.edu.au

\* Correspondence: Doupadi.herath@utas.edu.au

**Abstract:** This paper considers the feasibility of using augmented reality (AR) as a tool for enhancing visualization in maritime operations to avoid collision in different environmental conditions. According to the International Maritime Organization (IMO 2010), 90% of maritime accidents due to collisions at sea are caused in part by human error. This study investigates the new technology (AR) used to superimpose holographic images onto the real world; now reaching a state of readiness for commercial application. This paper demonstrates the competence of AR technology to serve as a maritime navigational aid. The research explores the viability of improving navigational safety in low visibility by projecting holograms of real-world objects in the same geo-location as the real object to make them "visible". The paper presents the logical deconstruction of the technical problems and identified solutions, together with results of experiments used to validate the concept and technology readiness for real word maritime application. The paper presents a verified demonstrator; a proposed holographic bridge interface with an innovative way of presenting information using AR technology. Furthermore, it identifies that new technologies offer the opportunity for enhanced operator performances, with the expectation being that this should lead to reduce risk to persons, property, and the environment.

**Keywords:** augmented reality (AR); compromised visibility; Microsoft HoloLens; navigation lights; virtual objects

## 1. Introduction

Recent decades have seen the appearance of a plethora of digital technologies on the ship bridge. Many are developed to assist navigation and collision avoidance in poor visibility, but presented on a range of screens and different formats, it is possible that operators may experience information overload. Navigation lights, for night operation, but not visible in fog condition have evolved over centuries and communicate relevant information to the seafarer in a very efficient way. The aim of this study is to evaluate the technical viability of using augmented reality navigation lights that are placed at the physical location of real obstacle to make them 'visible' in poor visibility conditions. The objective is to communicate critical navigational information to the seafarer in an effective and intuitive way, thus reducing the possibility of cognitive load.

To address the task this paper will first review the navigation data, coming from various navigation aids and examine them under different categories. The categories are capacity of information; priority rank of the available information; type of information process; extraction formats such as direct extraction methods, human interaction needed for data process, data extraction from other software. Then the identified data will be projected via holographic objects using software packages including Google SketchUp; simulations with Microsoft HoloLens; Bridge Simulator experiments to

develop exploratory case-studies. An example with 'foggy condition operations' is used to evaluate the impact on operational performance of the technology limitations and reliability when applied to the Augmented Reality (AR) platform. This paper presents a verified demonstrator for seafarers using their common language 'navigation light'; a proposed holographic object that can be projected in an innovative way of presenting information using AR technology. Furthermore, it identifies that new technologies offer the opportunity for enhanced operator performances, with the expectation being that this should lead to reduce risk to persons, property, and the environment.

To address the aims and objectives this paper will:

- Review the technology and its application as a maritime navigation aid;
- Identify the technical and practical limitations and opportunities with the chosen proprietary technology Microsoft HoloLens and select a practical solution;
- Validate and calibrate the chosen optimal solution using a dedicated hologram experimentation case study;
- Map out the functional opportunities presented by the findings.

The paper is organized as follows. Section 2 presents a brief introduction on background information about AR technology and current problems in ship operations at sea. Section 3 discusses how the AR can be used as a tool to overcome said problems in a more inventive way. Section 4 provides the AR aided visualization illustrated using artificial navigational light objects. Experimentation and field work of the study and a discussion on the results are presented in Section 5 followed by the conclusions presented in Section 6. Section 7 presents how to integrate the identified data coming from various navigation aids into a common platform using AR technology.

## 2. Background Information

The maritime traffic environment has consistently seen an ongoing increase in traffic and trends towards larger and faster ships, with the consequent increase in the risk of maritime accidents [1,2]. This could result in threats to the safety of seafarers, the economic performance of shipping industries, and the environment. Aiming to address this issue, enormous efforts have been made to develop measures to improve maritime safety, through improving technology involved in ship building and ship management, advanced navigation technology, and better crew training. International organizations such as the International Maritime Organization (IMO) and national Maritime Safety Administrations (MSAs) have published extensive rules and regulations on safety standards on the safe operations at sea. It is noted that most of the maritime accidents were due to collisions, contacts, and groundings [2–8]; making it important to try to mitigate factors that may cause these accidents. Furthermore, accident analyses have shown that human error is a dominant factor [9–11].

Over the last few decades, ship navigation and safety operations continue to push the limits of existing technology to overcome the current navigation problems using conventional technologies. With the development of new navigation equipment with different technology for safe navigation, the reduction of maritime accidents has not occurred as expected. It is postulated that one aspect may be due to the large amount of information provided to the seafarer on the ship bridge. Running into a 'big data' problem, the ever-increasing volume of information that is also not well organized may result in information overload for the operator. The information is displayed on a variety of different screens and monitors and to read it, the crew members must turn their attention away from the surrounding physical environment. This becomes even more difficult when a ship enters or leaves a harbor. Navigation may become an increasingly difficult task due to obstacles in the proximity of the ship, especially in different environmental conditions. Issues such as transfer of control, communication errors, lack of training, mechanical failure, and poor procedures may be exacerbated [12]. This becomes even more difficult with low visibility due to low light, fog, snow conditions, and rough sea conditions.

These findings show that it is important to use new technology to represent the navigation information more effectively to facilitate understanding and enable more accurate decision making for seafarers.

The Latin word "Augere" has the meaning of "to increase" or "to add", which is used to create the word "Augmented". Virtual reality (VR) visually immerses the user totally in a digital view. The new technologies that can be utilized to present information in a holographic form are reaching a state of technology readiness for possible application in supporting maritime operations and training. Therefore, it is important to investigate how AR may be employed to provide a complementary technology for enhanced safety in maritime operations in compromised visibility environments. If maritime accidents are to be reduced, it is important to consider if the information provided to crew is adequate for their decision-making in critical situations. It is also crucial to consider that the information provided to them and the way it is provided is making best use of available and emerging technologies.

The understanding of the navigation light operations at sea is a very familiar language for seafarers. Introducing new and different techniques however may cause more cognitive load. This paper will explore the common language of seafarers "navigation lights" and integrate it into the AR platform using the chosen AR hardware platform. This AR based new concept can be achieved using the international standard navigation lights as the holographic 'language' for representing objects in compromised visibility conditions. The idea of using navigation light to make objects more clearly visible is already known to seafarers; in fact, it is part of their cadet training under International Regulations for Preventing Collisions at sea (COLREGs)—Part C—Rules 20–31. Therefore, it is assumed that using AR navigation lights imposes no additional cognitive load for seafarers. Microsoft HoloLens include a Central Processing Unit (CPU) and Graphical Processing Unit (GPU), and the unit features a Holographic Processing Unit (HPU) that is responsible for the processing that integrates the real world and digitalized holographic data. It is important, therefore, to reduce the highly detailed projections to overcome the limitations on processing power and storage capabilities. To overcome this problem, this paper will focus on using lights instead of images of the whole ship or light house. This will increase the effectiveness of the headset for its users, allowing it to produce higher frame per second, thus reducing latency, lag, and delay [13].

When using AR, the user can still fully see the real world around them; with the addition of digital holographic objects as shown in Figure 1a. This figure is considered to be a daytime ship operation at sea and is difficult to visualize in compromised visibility conditions at night or in fog as illustrated in Figure 1b,c. Figure 1b,d shows a representation of the real-world situation, in that the navigation lights are visible at night (b) but not visible in fog (c). Figure 1d shows as an example that there is a possibility to overcome the problem by using augmented reality navigation lights; locked to the position of the known object in this case a holographic ship but in practice a real one.

AR has a history dating back to the 1950s [14], with a range of ever evolving proprietary platforms coming into the market in recent years. A literature review of the history and more recent contemporary technologies is provided in [15]. From this review, the Microsoft HoloLens was ultimately selected based on both its functionality and on account of being readily available. The unit used for the study is a first generation Microsoft HoloLens with the full specifications in [16].

Over the past few years, AR technology has become an interesting topic in various industries such as training and education, medical, engineering, architecture, defense operations [17–21]. The common AR device called Head-Up-Display (HUD) is being used to enhance the capability of pilots and reduce their time in fighter jets and commercial aircrafts [22]. In terms of developments lending to applying AR to maritime navigation, there are a number of studies published in the open literature.

Holder and Pecota [23] postulate that the 'head-down' position when using electronic navigation aids may make individuals reluctant to use them. These results suggested that there are some standard information requirements across situations that could be augmented with task and vessel specific information.

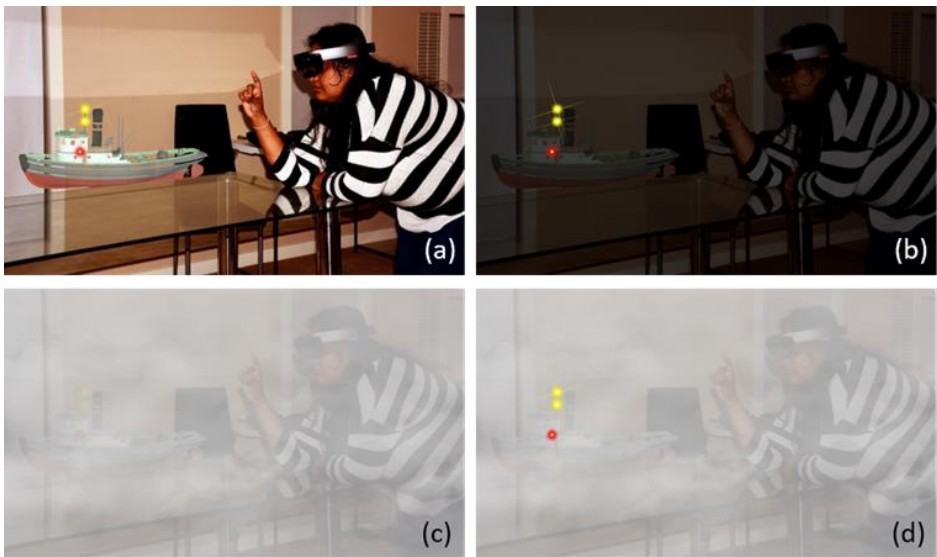

**Figure 1.** The user sees the real world as well as the virtual world (Augmented Reality (AR) vessel with navigation lights) in (**a**) normal condition; (**b**) night condition with lights visible; (**c**) fog condition where lights are not visible; (**d**) fog condition with holographic lights (giving a demonstration/visualization of concept).

Hugues, et al. [24] integrated a vision system with a thermal camera together with augmentation. Their study finds and concludes that the functionalities provided by AR must differ according to people and weather conditions, requiring a need to provide contextual information.

Oh, et al. [25] investigate 'Advanced Navigation Aids Systems based on AR'. They present work implementing the representation of information using AR; including conducting a survey of the outcomes. In their conclusion they find that, "*Although various types of navigation equipment with advanced technology are available to help enhance operation safety for navigators, further research on the method of efficiently displaying and servicing information rather than indiscriminately providing a bounty of information is necessary*".

Radu [26] explores the use of AR in education, finding both positive and negative effects on learning. Further, Wu, et al. [27] argue that, from an educational perspective, AR should be viewed as a concept, rather than a technology.

AR can project the information from display devices in bridge equipment to appear as a head-up display. Using a more advanced and contemporary application, this paper proposes that AR can also be used to create holographic representations of real or enhanced objects. In this way, complex information from multiple sources, may be presented to the viewer as if it were part of the real world such as seeing a real/virtual object coming towards you; rather than needing to interpret information from a digital display.

*Deconstruction of the Technical Problem (Research Problem)*

The safety of passengers and vessel is one of the major concerns in maritime industry. Despite advances in navigational technology, and improvements in human, legal, and social aspects, ship accidents continue to occur. Of all the maritime accidents reported to date, the highest number of accidents that occurred are due to errors during operations. Among these accidents, ship collision has a high portion and has drawn much attention from the public [2]. Moreover, these accidents have been caused by human error due to inadequate watch keeping and mistakes in ship handling [25]. Therefore, it is important to understand the factors that contribute to these accidents and implement the correct actions to prevent these accidents as soon as possible.

The bridge is the brain of the ship and in recent years has become more modernized; with advanced equipment and tools. It is important for bridge crew members to have an extensive understanding of bridge systems; in addition to navigation skills [28]. There is a large amount of information available on a modern ship bridge, and not all of that information is required for the navigator. In fact, Hareide [29] notes that it is important for the navigator to sort out the information for planning a safe navigation path and that this is difficult when attention is divided between digital displays and the outside world.

As a physical computing and display technology, AR devices are quickly becoming widely used in many industries. This wide adoption implies that it cannot be reasonably argued that the technology is totally useless. Nevertheless, it is only a tool and it is 'what it is used for' that both provides value and provides the foundation of this research. Addressing the contemporary problems associated with 'big-data', the very large amount of data given to the seafarer in a variety of different forms and formats as discussed earlier, this research proposes to investigate 'how' the AR platform can be exploited for organizing and presenting bridge information. Though the aspiration would always be that better organized information should lead to better performance and thus enhanced safety, this will remain implicit within this study. This research paper develops the initial requirements for expected functionality in the final solution and formally address the verification and validation of this functionality at the technical level. It therefore strives to provide a solution for a situation where seafarers may struggle to cope with the very large amount of information provided by the current digital displays. It aims to improve the safety, through collision avoidance, in compromised visibility by organizing and rationalizing information and by exploring 'how' to present that data using a holographic object but with existing data technology. In particular, the integration of data in later stage from growing sources and visualization of that data can improve the safety of navigation in the following aspects:

1. Visualization of normally nonvisible or not clearly visible risky objects;
2. Visualization of risky patterns that can only be captured by expert navigators and;
3. Facilitate quick processing and visualization of information that are distributed in multiple sources which require extensive manual processing.

By making it possible for the seafarer to make navigational decisions based on their view of real and virtual objects, rather than having to interoperate digital display information, it is hoped that the process will become more intuitive for the user. Furthermore, it identifies that new technologies offer the opportunity for possible enhanced operator performances, with the expectation being that this should lead to reduction of risk to persons, property, and the environment.

The main reason to represent holographic objects in terms of navigation light operations at sea is due to the common and familiar language for the seafarers. Furthermore, holographic navigation lights have the power to communicate most of the visual clues that disappear under compromised visibility environment conditions discussed in the next section.

## 3. AR as a Viable Navigation Aid in the Maritime Industry

Navigators on modern ships work in a technologically advanced environment compared to older times. Therefore, it is important to understand how to identify/distinguish important data from different data sources for example but not limited to: Electronic Chart Display Information System (ECDIS), Radar, sonar/depth-sounder integrate into the AR platform in a more intuitive way to improve operations in compromised visibility, while addressing the 'big-data' problem. This problem is commonly known as the very large amount of data given to the seafarer in a variety of different forms and formats.

### 3.1. Review of Ambient Data Sources

It is important to understand the ambient information available to the seafarer. This may include visual clues apart from the navigation aids, and the conscious or unconscious usage of the information

during operations at sea for safe navigation. It is postulated here that these visual clues may disappear in compromised visibility conditions. To address this possibility, this study proposes to review and critique ambient information sources; for possible inclusion in an AR model. Table 1 represents the identified visual clues used in vessel navigation in daylight. Table 1 also strives to suggest/offer possible ways of replacing the 'clues' in an AR platform.

**Table 1.** Identifying objects in night environment conditions.

| Navigation Aid Type: | Fixed Features | |
|---|---|---|
| **Information:** | **Example Feature/Object:** | |
| Marked (lit) land feature, lighthouse, leading line marks | **Lighthouse**—fixed light of known position and height, identifiable by unique light flashing pattern (attributed to that particular lighthouse). May include colored light sectors (red, white, green) indicating approximate position line (bearing to mark). | |
| **Transmutation:** | | |
| ECN, RADAR, (AIS) | **Leading Marks**—usually display a triangle shape. The front lead has its apex pointing up and the rear or back lead has its apex pointing down. When leads in line, observer (vessel) is in middle of channel. At night, major leads are usually lit with blue lights. **Blue Middle Channel Marks**—fixed blue lights that mark the middle of the channel for vessels passing under a bridge. Directional and sector lights have a similar purpose to leading lights at channel entrances and inshore waters. | |
| **Leads to know:** | | |
| Position line with respect to fixed object(s) | | |
| **Suggested AR indicator** | | |
| Int. Standard navigation light (light, color, flashing pattern) | **Directional Lights**—used as steering marks and sector lights display a light of different colors (usually green, white, and red). **Port Traffic Signal Lights**—three red vertical flashing lights that are remotely controlled by Maritime Safety traffic services to tell others in the area that large commercial ships are moving in the port, harbor, marina, or other confined waterway. Vessels must not enter or depart the port or harbor area when the port traffic signal lights are flashing. | |
| **Navigation Aid Type:** | **Buoys, Marks, and Beacons** | |
| **Information:** | **Example Feature/Object:** | |
| Waterway floating device system display information need for navigation at day-time and night-time | **Lateral Marks**—show the port (left) and starboard (right) sides of navigable waters or channels. A port mark is red with a can-like shape. A starboard mark is green with a cone-like shape. At night, a port buoy shows a red flashing light and a starboard buoy shows a green light. **Cardinal Marks**—show where the deepest and safest water to travel is. Marks have black and yellow bands with black double cones on top showing the different compass direction that identifies the safest and deepest water to navigate according to each direction (North, East, South, and West). At night, each type of cardinal mark has a flashing white light with different groupings of flashes (continuous, or groups of 3, 6, or 9). | 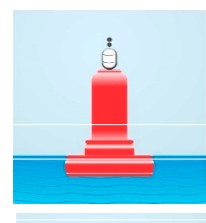 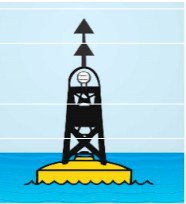 |
| **Transmutation:** | | |
| ECN, RADAR, COMPASS | | |
| **Leads to know:** | | |
| Position line with respect to fixed object(s) | | |
| **Suggested AR indicator** | | |
| Int. Standard navigation light (light, color, flashing pattern) | | |

**Table 1.** *Cont.*

| Navigation Aid Type: | Buoys, Marks, and Beacons | |
|---|---|---|
| Information: | Example Feature/Object: | |
| | **Isolated Danger Marks**—show where there is an isolated danger that has navigable water all around it (for example, an isolated shoal, rock, or wreck). Marks are black with one or more red horizontal bands and two spheres as the top mark. At night, the white light flashes in groups of two.<br>**Safe Water Marks**—show that there is navigable water all around the mark. For example, fairway, mid-channel, or landfall mark. Marks have red and white vertical stripes with a single red sphere as the top mark. At night, a single white light shows one long flash every 10 s.<br>**Special Marks**—show a special area or feature. For example, to show that a channel divides or to mark cables or pipelines. Special marks are yellow and sometimes have a yellow X as the top mark. At night, the flashing light is yellow.<br>**Emergency Wreck Marking Buoys**—used to identify new dangers or wrecks. They have blue and yellow vertical stripes and are a pillar or spar shape with a yellow cross as the top mark. At night, the flashing light alternates between 1 s of blue light and 1 s of yellow light, with 0.5 s of darkness in between. | 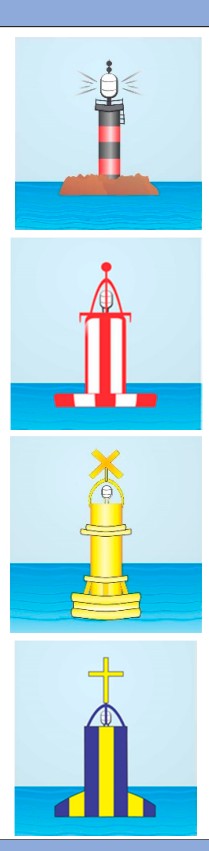 |

| Navigation Aid Type: | Identify Speed through Water | |
|---|---|---|
| Information: | Example Feature/Object: | |
| Speed through water and depth of the water | **Speed through Water**—speed through water and speed over the ground are independent. Watching water near the bank going by may give one impression whereas watching the flow around a buoy, post, or rock may show that the speed through water is different.<br>**Speed/Depth through water**—the characteristic Kelvin wake of a ship changes angle in shallow water, and is sometimes observed by the pilot to indicate water depth. | |
| **Transmutation:** | | |
| Visual | | |
| **Leads to know:** | | |
| Speed or depth of the water with respect to fixed object(s) | | |

A conclusion that can be derived from Table 1 is the importance of the visual clues apart from the navigation aids during operations at sea for safe navigation such as seeing the water flow around a buoy or bank. Furthermore, it has been shown that the disappearance of these visual clues at night can lead to maritime accidents which damage human lives, property, and environment. It is therefore very important to develop a new concept of safe navigation with the help of AR technology to enhance visualization of seafarers in compromised visibility conditions. This can be achieved while presenting lots of visual clues information in the form of holographic objects in the AR platform at compromised visibility conditions such as night, fog, or snow. This paper evaluates the technical viability of using navigation lights as AR objects which are geo-locked to the real object, and the capability to represent information that is normally unavailable in poor visibility.

### 3.2. Navigation Light Requirements

According to the IMO, COLREGs, navigation lights should appear steady and non-flashing. The COLREGs specify the required number, location, position, color, luminosity, number of spare lights, of navigation lights that must be present on different vessel types and in different operations [30]. Therefore, it is important to understand the regulations on COLREGs for developing holographic objects in an AR environment to symbolize the real light intensity in a reasonable and realistic way.

## 4. AR Aided Visualization Illustrated Using Artificial Navigational Light Objects

### 4.1. Derivation of Holographic Signals

Having identified the international standards for navigation lights, the task of identifying a suitable holographic display is undertaken. The aim of this section is to determine the method of holographic navigation light display and calibrate the derived objects lights. The objective is to define the baseline object light that is meaningfully displayed to a degree of realism for further development of the platform case study. This will be addressed in the following section by, first, presenting the approach/method for calibrating the holographic signals; second, presenting the model design and experimental methodology, then presenting the results and discussion.

### 4.2. Calibration of Holographic Signal

It is first important to note that the HoloLens technology is designed and calibrated to operate in the near environment such as with a physical room. To verify the functionality and calibrate the outputs on typical scales of kilometers, case studies were used. A case study was conducted to examine how holographic navigation lights may be used for visual awareness of objects/obstacles when in compromised visibility. Holographic spheres are used to represent different virtual lights, for vessels at different distances. The experiment used a sphere as a 'virtual light', which represented the other vessel's navigation light to the seafarers as the view is the same from all directions and is just a way of modeling 'brightness' in a practical way. The main purpose is to place a hologram 'light' in the same place as a navigational light on another vessel so that it appears as it would at night but in the intended case that it will appear this way on a foggy day to aid the user to intuitively account for the object with the least cognitive load.

Graphical software was used to generate arbitrary models of navigation lights, and to compare their holographic appearance at different distances with real lights. The SketchUp software was used for the modeling as it has a readily usable interface for the HoloLens. Relevant tools that are available in the software were used to prepare models and then projected using a head-mounted holographic visualization tool in a Microsoft HoloLens device.

### 4.3. Model Design and Experiment Methodology

The study considers spherical virtual lights of different radii (0.5, 0.75, 1, 2, 3, 4, 5, 6, 7, 8, 9, 10 m). The different radii are explored in this case to simulate a visual approximation of luminosity of the holographic light. A model was developed and subsequently used to identify and calibrate the luminosity of the virtual navigation light at different distances.

The images of different light luminosity were calibrated for eight sets, representing eight selected distances (150, 500, 1000, 1500, 2000, 2400, 2800, 3600 m) covering a variety of terrain scenarios including land, open water, and various background lights.

The designed model projected into the Microsoft HoloLens Sketchup Viewer through Google SketchUp "SketchUp Viewer AR/VR" uses the same Wi-Fi network on both devices. The projected model is then displayed on a 2.3 Megapixel widescreen stereoscopic head-mounted display.

The Microsoft HoloLens is the first commercially available AR Head-Mounted Device (HMD) to reach the market as a consumer version which was first released as a development release in 2016. Since this device is so new, and the HoloLens is the first to market in this area, there are little to no

competitors for consumer grade wireless AR HMDs. The Microsoft HoloLens features four Intel Atom x5-Z8100 1.04 GHz Intel Airmont Logical Processors, an HPU/GPU Holographic Processing Unit, 64 GB Flash, 2 GB RAM, 2.3 Megapixel widescreen stereoscopic head-mounted display. HoloLens runs with Windows 10 operating system. Holograms enable the user to apply five fundamental features of HoloLens which are gaze control, gesture control, voice controls, spatial mapping, and spatial sound.

Using the defined 3-D models, projected through the holographic interface, the visual appearance of the virtual lights was compared with real lights of known distance and brightness. The images were captured and used to assess the 'best-fit' solution.

## 5. Experimentation and Field Work

### 5.1. Initial Field Work

Before moving into the results of the case study, it is important to look into how a projected holographic boat/ship appears when viewed from the bridge of a real ship. The objective was to see how the HoloLens technology and derived holograms look and function in the real environment and to identify any practical/technical issues that may not be apparent from desk studies.

In this field study, a holographic fishing boat is examined, when viewed from the bridge of the vessel Bluefin (Australian Maritime College Research Vessel) as illustrated in Figures 2 and 3. The holographic boat highlighted in Figure 2 by the white arrow is found to be fully visible in daylight conditions. In addition, it was possible to make the holographic image appear to be outside the ship bridge; despite being viewed through the glass windows on the bridge.

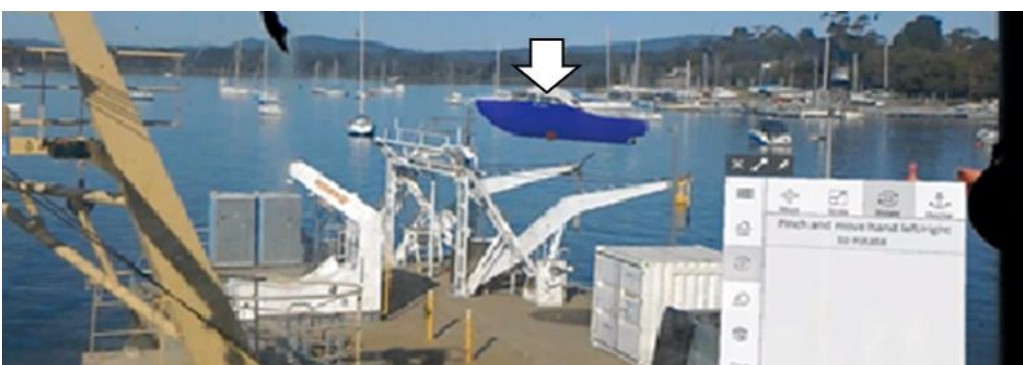

**Figure 2.** Projected view of fishing boat at AMC Surface Vessel Bluefin at angle of 115.4°.

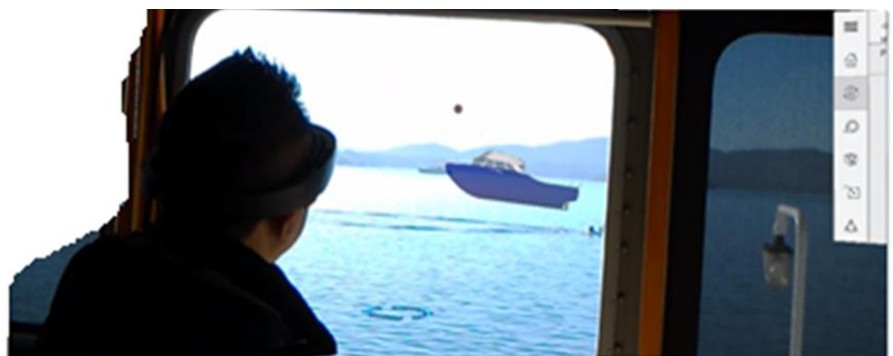

**Figure 3.** Projected view of fishing boat at AMC Surface Vessel Bluefin at angle of 75.6°.

Figure 3 shows the same holographic model being viewed from a different location on the ship bridge including a second viewer, wearing a HoloLens. It is worth noting that the technology used is capable of pairing, so that multiple bridge personnel can view the same virtual object at the same time. This may help the bridge crew member's perception, especially in confirmation bias which can

be a serious threat to safety. The confirmation biasing is when the crew members only understand the information which can confirm their assumption and reject any information that falsifies their assumption. The observations from Figures 2 and 3 show the exploration of holographic objects with a real world environment. These results are used as proof-of-concept; that AR technology can be implemented for maritime industry to enhance visualization of the operations at sea.

From the field experiments it was identified that projecting images of real ships added 'visual clutter' that was not information rich. Additionally, projecting the hologram of a full ship uses a significant amount of CPU availability. By deconstructing the problem, and in discussion with seafaring experts/instructors, the processes of projecting only navigation lights were identified. The justification for this is based on the assumption that seafarers are already familiar with navigation lights as a 'language'; resulting in less computation time for the device and an information rich output.

*5.2. Field Studies for Navigation Lights*

The aim of this experiment was to demonstrate the view of the projected holographic object appearance when viewed in night vision, especially displaying navigation light. The holographic technology is typically calibrated for indoor use, so the second aim of the experiment was to identify if the technology functioned adequately in a real outdoor environment; on scales of hundreds of meters rather just a few meters. The objective was to calibrate the visual objects so that they appear as close to reality in terms of size and luminosity as is practicable. There are two sets of surveys that were conducted: one in Newnham Tasmania, Australia and one at Lake St. Clare Tasmania, Australia. The first stage occurred during the first stage of COVID-19 lockdown; requiring some lateral thinking. To progress, a practical solution was identified, by calibrating against a streetlight at a known distance and the outcomes are presented in Figure 4. The result demonstrates that the virtual object can be created in an outdoor environment. The distance between the street-light pole and operator or observer is 150 m and the holographic objects are placed in the same distance from the operator or observer. The same experiment was conducted using different colors of spheres such as red, green, and blue as shown in Figure 5; which are commonly used in navigation lights.

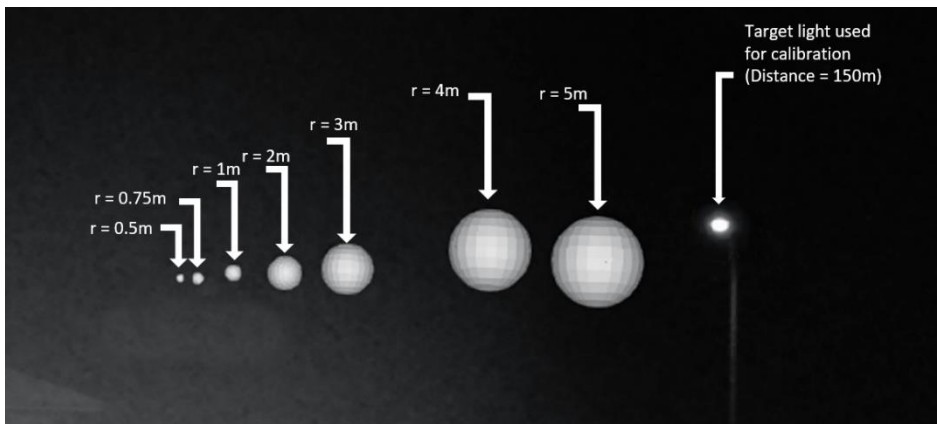

**Figure 4.** Matching spheres (size and luminosity) with the actual light at a distance of 150 m.

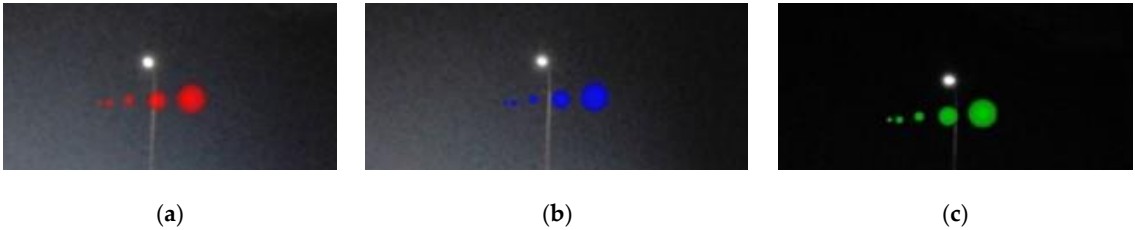

| (a) | (b) | (c) |

**Figure 5.** Matching spheres of different colors (size and luminosity) with the actual light at a distance of 150 m. (**a**) Red color spheres; (**b**) blue color spheres; (**c**) green color spheres.

The next set of experiments were conducted in Newnham Tasmania, at 2400, 2800, and 3600 m across the Tamar River. Figure 6 gives the example with a 2400 m distance.

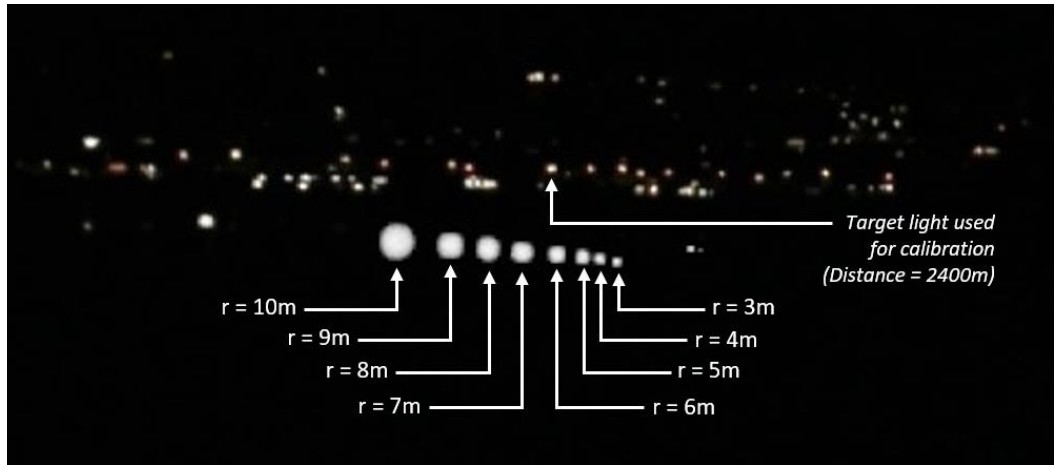

**Figure 6.** Comparison of light and sphere at a distance over water of 2400 m in Newnham Tasmania.

The survey was conducted in Newnham across Tamar River, a metropolitan area with lots of lights. This made it difficult to determine the light reflection of a single light on the water. To add an additional reference point, and using more controlled conditions, extra tests were conducted in a rural area with very low light pollution, which is assumed to be more representative of an ocean environment. This was undertaken at Lake St. Clair beach where a pump house is located 1000 m away from the beach. The same experiment was conducted, using the same model; set to 1000 m as shown in Figure 7.

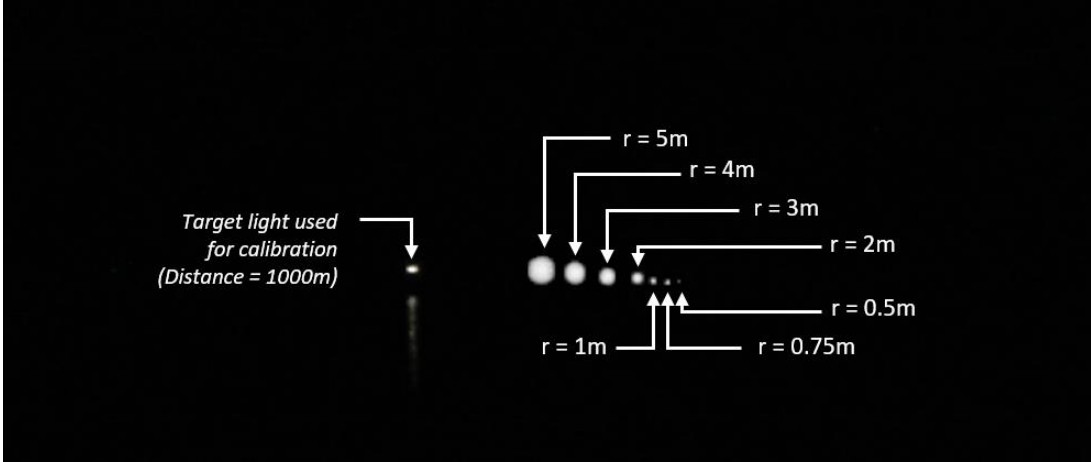

**Figure 7.** Comparison of light and spheres at a distance over water of 1000 m (Lake St Clair).

Figure 7 shows the navigation light projection visualization on the river or sea with the reflection on the water. This image provided an additional concept information that will be considered at a later stage of the research.

The results presented in this section show the exploration of holographic objects in a real-world environment. These results provide proof-of-concept, that AR technology can be implemented for projecting virtual navigation lights that resemble visual appearance and luminosity at a distance between 150 and 3600 m in a dark environment. These results were conducted using a small datasets during the night. Ongoing research will investigate visual appearance of the derived holograms in other light conditions, especially in fog condition. The final outcomes of the proposed augmented

reality navigation light can be visualized through the physical window as shown in Figure 8. This result showed that without the redundancy of information in terms of without having full shape of the object, it is still possible to give the proper dimensions and the direction of the ship correctly. Moreover, this result revealed the enhancement of visualization of AR device Microsoft HoloLens for ship operations at night.

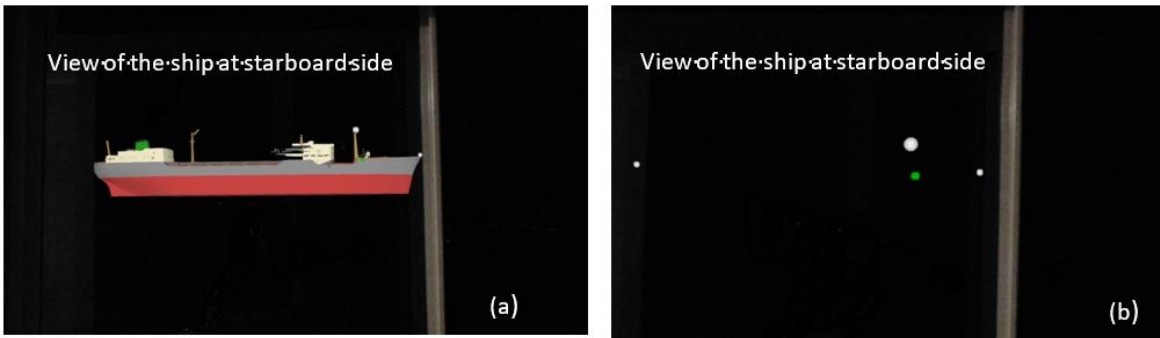

**Figure 8.** (**a**) The projected holographic ship through the physical window and (**b**) the navigation light projection of the same ship through the ship bridge.

### 5.3. Data Process and Data Analysis

Data is collected by projecting holographic lights at various distances and comparing them to known lights at the same distance. To model the holographic lights, a set of spheres of different sizes are generated in a 3-D modeling software and imported into the holographic projection hardware. The holograms are then projected in a night environment over a body of water to lights of known size and distance. The calibration light used is standard type of street-light and is used throughout so that all samples are of the same size and lumen.

### 5.3.1. Data Processing

Images of the target light and closest fit holographic lights are captured using the same scale as shown in Figure 9, Row 1. To account for the dispersion of the light over the image and the fuzziness, an image processing software is used to convert the image into three distinct regions of brightness as illustrated in Figure 9, Row 2. The area of each of the three regions is then estimated by comparison with a 10-by-10 grid; giving a relative percentage for each region. From this an average is taken for each image and regression analysis is used to identify the best match size. A similar process was carried out for all the data sets used in both case studies to identify appropriate holographic spheres for best matching size to represent navigation lights in the real environment.

### 5.3.2. Data Analysis

The best match size is then plotted against distance and the relationship estimated as illustrated in Figure 10. The data is somewhat spread so there is no justification to use any higher order curve than a linear fit. This results in the estimated representative holographic radius, $r$, as a function of distance $D$, given by Equation (1):

$$r = 0.001D + 0.4308 \tag{1}$$

with a standard regression of 0.7. Standard experimental uncertainty methods are used to identify the variable uncertainty and sensitivity of each input and are found to give a combined uncertainty of $+/-0.08$ m for the estimated radius.

Analysis of the results shown in Figure 10 confirms that holographic images of suitably sized spheres can reasonably be used to represent navigation lights in a real-world environment. The calibration process identifies that a simple linear relationship between sphere radius and distance from objects can be used to give a reasonable approximation to size and luminosity.

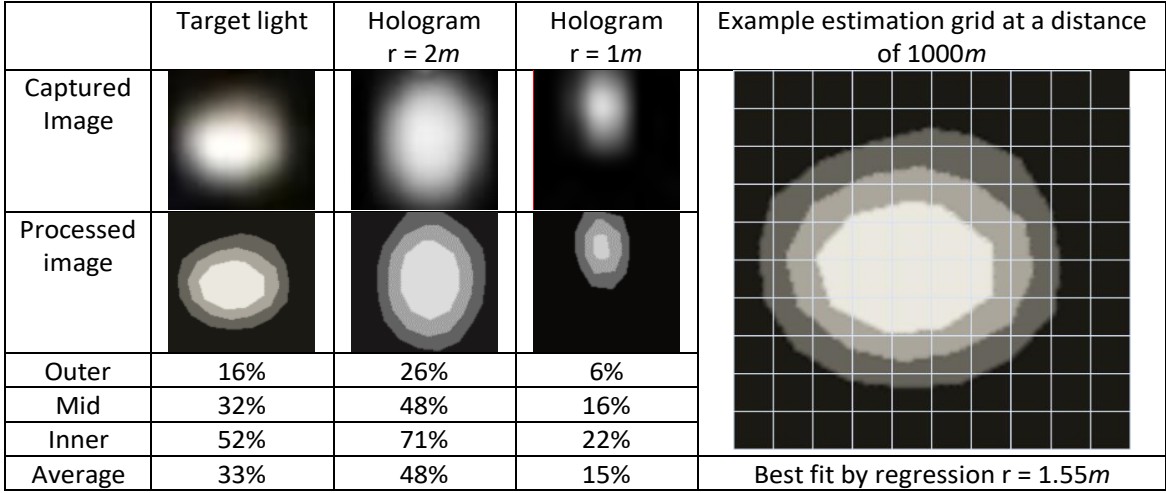

| | Target light | Hologram r = 2*m* | Hologram r = 1*m* | Example estimation grid at a distance of 1000*m* |
|---|---|---|---|---|
| Captured Image | | | | |
| Processed image | | | | |
| Outer | 16% | 26% | 6% | |
| Mid | 32% | 48% | 16% | |
| Inner | 52% | 71% | 22% | |
| Average | 33% | 48% | 15% | Best fit by regression r = 1.55*m* |

**Figure 9.** Example of identified best holographic spheres to represent navigation light in the real-world environment.

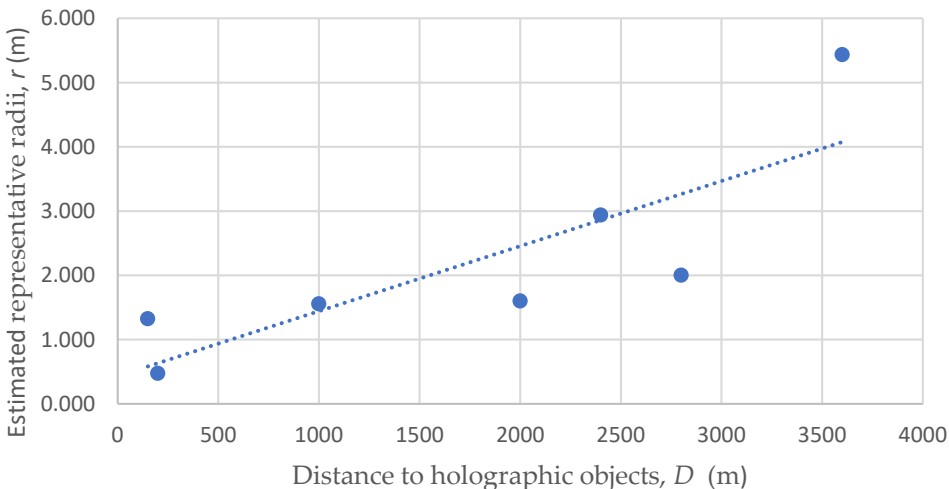

**Figure 10.** Linear regression estimation curve for the representative hologram (sphere) radii as a function of distance.

*5.4. Discussion*

On examination of the available information and processor capacity, navigation lights are identified as the most intuitive candidate for a holographic language. This conclusion depends on the efficiency of the processor which is used to project the objects for holographic display when compared to, for example, projecting the whole ship or lighthouse. In this paper it is postulated that seafarers are already familiar with navigation lights and as such would not have to learn a new skill.

This research paper has identified the possibility of using virtual lights to mimic the navigation lights in other vessels and operations at sea in compromised visibility conditions. Proof-of-concept is provided that holographic navigation lights can be achieved with existing holographic technologies and in environments including distance over water comparable to those in real navigational situations.

Other visual clues such as light reflection in the water are noted. Their importance for realism such as cognitive acceptance will be considered, and outcome will inform future representation of AR objects.

This research paper contributes to the area of Holographic AR based new algorithm for advanced ship navigation, safety, and operations at sea using Microsoft HoloLens for the first time in the maritime industry and for maritime operators. This ongoing research proposed to develop further the

concept in vision-based navigation using AR as a tool for enhancing information propagation; with the expectation of improved situational awareness. The next stage of the study will formalize the process for identifying and extracting information such as location of other navigational hazards. This will then be used to inform the geo-location of the AR navigation light presented in this paper.

## 6. Conclusions

This paper outlined the viability of representing AR holographic navigation lights for ship operations to enhance the visualization of navigational hazards; in terms of safety in compromised visibility. It is identified that there is a significant amount of research work being conducted to realize safety operations at sea, and that safety operations at sea is the major challenge in maritime industry especially in compromised visibility. It is further identified that AR is reaching a stage of technology readiness suitable for commercial application. Holographic navigation lights that are geo-fixed at the location of real objects are identified as an information rich way of presenting hazards. Field studies are conducted and demonstrate the functionality of an AR technology platform in a real maritime environment. Moving beyond the calibrated and intended scope of the device used, a practical way of representing realistic lights is derived and verified at representative distances in km scales. View size and luminosity of the AR objects are calibrated using regression analysis, and a suitable scaling factor identified.

The study identifies that using AR navigation lights that could be geo-locked to real physical hazards is a viable way of presenting navigational information in compromised visibility.

## 7. Future Work

Future work will deliver a real-time AR integrated and interactive user interface to demonstrate the advantages of using AR technology while operating in compromised visibility such as snow, fog, rain, and rough seas. This can be done by developing a new algorithm to import data into the AR platform to identify objects in different light and environment conditions and use it to run the application as illustrated in Figure 11. Future research aims to:

- Identify the key data sources, their purpose, and their data output format;
- Identify ways to extract the necessary data;
- Unify the various data and data protocols into an integrated platform;
- Develop a case study in AR demonstrator platform.

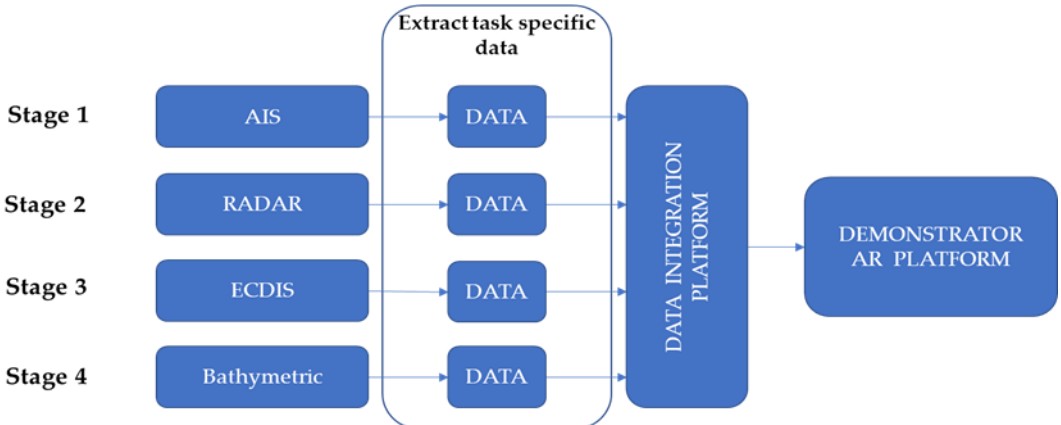

**Figure 11.** Overall design process.

Though this study has a focus on the technology, it is also important to recognize the need to verify the user interface using human subjects. It is recommended that future studies explore the

response and opinion of seafarers; who will ultimately use the technology. Future work would also strongly recommended field research with examples given by Procee et al. [31,32].

**Author Contributions:** D.B.—PhD researcher; M.W.—Primary supervisor; C.C. and D.J.—Co-supervisors. All authors have read and agreed to the published version of the manuscript.

**Funding:** This research received no external funding.

**Acknowledgments:** Our appreciation and special thanks go to Damien Freeman for his great support and guidance. We also would like to thank all the staff members in the bridge simulator, AMC and University of Tasmania.

**Conflicts of Interest:** The authors declare no conflict of interest.

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
