# Peer review of "Augmented Reality Lights for Compromised Visibility Navigation"

_jmse, doi:10.3390/jmse8121014_

Round 1
Reviewer 1 Report
The presented research addresses a very important issue and provides an interesting solution. The authors may consider mentioning the aim of the paper only once, at the beginning of the paper. Also, it would be interesting to read are there plans to investigate seafarers` opinion.
Author Response
Dear Reviewer,
On behalf of my co-authors, I would like to thank you for your time on this manuscript. All the comments are carefully considered, and suggested changes are incorporated in the revised manuscript. Please refer the attached document for your reference.
Thank you.
Doupadi Bandara

Reviewer 2 Report
Dear Authors,
The topic is very interesting and promising.
Below please find comments to improve scientific soundness.
- starting abstract with a question is more a conference approach.
- in line 11 you mentioned factor 90%, in line 76 you mention factor 80%... You don't know or you guessing? I suggest finding the data from the year 2020 and write the exact number of collisions caused by human error and add the source in the reference.
- I strongly suggest avoiding brackets in the text
- Explain what you want to tell in lines 75 - 81. How these facts directly concern your research? Rephrase it. Is it just to make ballast in text or do you want to tell something?
- line 107 - avoid "cognitive workload". Use "cognitive load" or "workload"
- line 111 - why you imply that "concept is already known to seafarers"? some reference perhaps?
- line 120-121 goes to the top of the section or into the introduction
- line 134 - reference missing
- line 136 - self-citation
- line 168 - "therefore less demanding" is speculation. Rephrase it.
- line 169-171 - goes to the introduction
- 186-187 - out of context
- 199 - how do you define "cognitively overloaded environments?
- 209-211 - already mentioned in the introduction
- 223 - explain "intuitive way"
- 291 - 239 rephrase
- 337-339. You already described the aim in the introduction. So I suggest to rephrase it
- 378 - which results?
- 383 - Strongly suggest to make some field research. Perhaps include in the references also Stephen Procee: Toward functional AR(2017), AR to improve collision avoidance(2018), etc...
- The overall experimental design is not clear,
- The results are weak
- Figure 11 is promising :)
The scientific soundness must be improved.
Best Regards,
Author Response

(The authors gave the same response as above.)

Round 2
Reviewer 2 Report
Dear Authors,
Thank you for accepting the suggestions.
The manuscript has been significantly improved.
Although: Too frequent use of the brackets in the text is not a reader-friendly (suggestion No.3).
Wish you all the Best,
Author Response
Dear Reviewer,
On behalf of my co-authors, I would like to thank you for your time on this manuscript. All the comments are carefully considered, and suggested changes are incorporated in the revised manuscript. Moreover, we would like to thank you for accepting our previous responses and providing further suggestions to improve our manuscript. Please refer to the attached document for your reference.
Thank you.
Doupadi Bandara
